# Improved Motion Artifact Correction in fNIRS Data by Combining Wavelet and Correlation-Based Signal Improvement

**DOI:** 10.3390/s23083979

**Published:** 2023-04-14

**Authors:** Hayder R. Al-Omairi, Sebastian Fudickar, Andreas Hein, Jochem W. Rieger

**Affiliations:** 1Applied Neurocognitive Psychology Lab, Carl von Ossietzky Universität Oldenburg, D-26129 Oldenburg, Germany; 2Department of Biomedical Engineering, University of Technology—Iraq, Baghdad 10066, Iraq; 3Assistance Systems and Medical Device Technology, Carl von Ossietzky Universität Oldenburg, D-26111 Oldenburg, Germany; sebastian.fudickar@uni-luebeck.de (S.F.); andreas.hein@informatik.uni-oldenburg.de (A.H.); 4Institute for Medical Informatics, University of Lübeck, D-23538 Lübeck, Germany; 5Cluster of Excellence Hearing4all, Carl von Ossietzky Universität Oldenburg, D-26129 Oldenburg, Germany

**Keywords:** functional near-infrared spectroscopy (fNIRS), real fNIRS data, motion artifact, motion correction

## Abstract

Functional near-infrared spectroscopy (fNIRS) is an optical non-invasive neuroimaging technique that allows participants to move relatively freely. However, head movements frequently cause optode movements relative to the head, leading to motion artifacts (MA) in the measured signal. Here, we propose an improved algorithmic approach for MA correction that combines wavelet and correlation-based signal improvement (WCBSI). We compare its MA correction accuracy to multiple established correction approaches (spline interpolation, spline-Savitzky–Golay filter, principal component analysis, targeted principal component analysis, robust locally weighted regression smoothing filter, wavelet filter, and correlation-based signal improvement) on real data. Therefore, we measured brain activity in 20 participants performing a hand-tapping task and simultaneously moving their head to produce MAs at different levels of severity. In order to obtain a “ground truth” brain activation, we added a condition in which only the tapping task was performed. We compared the MA correction performance among the algorithms on four predefined metrics (R, *RMSE*, *MAPE*, and Δ*AUC*) and ranked the performances. The suggested WCBSI algorithm was the only one exceeding average performance (*p* < 0.001), and it had the highest probability to be the best ranked algorithm (78.8% probability). Together, our results indicate that among all algorithms tested, our suggested WCBSI approach performed consistently favorably across all measures.

## 1. Introduction

Functional near-infrared spectroscopy (fNIRS) is a non-invasive neuroimaging technique that uses light in the near-infrared range to measure hemodynamic response changes associated with neuronal activity in the brain [1,2]. It exploits the fact that the scalp and skull are relatively transparent to light in the infrared range. For brain activation measurements, infrared light with wavelengths (e.g., 760 and 850 nm) is emitted from a source into the brain tissue. The light arriving at a detector placed several centimeters away is measured and converted to concentration changes of oxyhemoglobin (HbO) and deoxyhemoglobin (HbR) using the modified Beer–Lambert law [3]. The concentration changes are interpreted as neuronal activation changes in the superficial cortical layers because the received light was scattered along a bent path through the superficial brain tissues. This technique allows for more head movements than other imaging techniques, such as fMRI, as the optodes (sensor detector pairs) are typically attached to the head. This makes it a suitable method for brain research in relatively realistic or real-life settings, such as infant brain development [4,5], functional connectivity [6], cognitive science [7,8], psychiatry and neurology [9,10], anesthesia [11], and driver monitoring [12].

However, although fNIRS allows for head movements, its signal quality is affected by motion artifacts (MA). Any change in the orientation or distance of the sources and detectors relative to the skull caused by movements can change the impedance and generate changes in the fNIRS signal. For example, spikes occur when the optodes quickly move and return to their original position. In addition, slow signal drifts happen when the optodes are slowly moved, and baseline shifts occur when the optode location changes [13,14].

Because of the wide dynamic range of MAs, which overlaps with the dynamic range of the hemodynamic responses of the brain, MAs cannot be removed by simple time domain filtering without affecting the measured brain activation. Moreover, the MAs’ dynamics vary between participants, requiring individual filter settings [15]. Therefore, the first step in recording fNIRS is to reduce optode movement to avoid MAs as much as possible. For example, this can be done by fixating the optodes firmly in a support structure attached to the head (e.g., a cap) [16]. Once MAs are in the data, the rejection of measurements with MAs would be the simplest approach to deal with the problem. However, in many experiments, the total number of trials is rather small, and, therefore, data loss due to the rejection of trials is undesirable and can reduce the reliability of the results [17]. In these cases, correction of the MA contaminated signal is crucial to avoid data loss [3].

Several MA correction algorithms have been developed to prevent data loss. They follow different rationales and have different strengths and weaknesses. In multi-sensor recordings, principal component analysis (PCA) can be efficient. It exploits cross-correlations between sensor signals caused by movements of the whole sensor array. It decomposes the multi-sensor signal into a set of orthogonal spatial patterns of sensor activations, in which, ideally, the MA segments are separated from the brain signal [17]. Unfortunately, PCA applied to the whole measured signal often does not separate brain and MA signals well and tends to over-correct the signal [18]. Targeted principal component analysis (tPCA) is an extension of classic PCA designed to avoid over-correction. Therefore, it focuses only on the segments with MA to obtain a better separation of MA and brain signal, and to correct only those epochs with identified MAs. However, the method is complex to use as its performance depends on a number of parameters the user has to set [19]. The correlation-based signal improvement (CBSI) method assumes that brain-generated HbO and HbR signals are always negatively correlated, whereas intervals with MAs are indicated by positive correlations [20]. Consistent negative correlation is a strong assumption that is needed to estimate the HbR from the HbO signal in CBSI. However, it may sometimes not be met, for example in some pathophysiological conditions [21]. CBSI is known to effectively remove large spikes and baseline shifts. Another advantage is that it can be fully automated [20]. While the previously mentioned motion correction algorithms exploit correlations between time series, other MA correction methods operate on single channels. The movement artifact reduction algorithm (MARA) [14] combines MA detection with spline fitting to subtract an estimate of the MA. Similar to tPCA, MARA depends on MA interval detection. This complicates the application as many user parameters need to be selected. Conversely, Savitzky–Golay (SG) filtering [22] locally fits a polynomial function to estimate the signal without MA, and replaces the measured value with the estimate from the fitted polynomial. Robust locally estimated scatterplot smoothing (RLOESS) [23] implements a smoothing filter by initially fitting a locally weighted regression, which is iteratively refined based on the size of the residuals. Both RLOESS and SG are denoising filters that can remove high-frequency MAs but are relatively ineffective in removing slow drifts and baseline shifts. In addition, RLOESS is computationally expensive [24]. In yet another approach, wavelet filtering [25], the measured signal is first decomposed on a wavelet basis; then, the wavelet coefficients considered to represent an artifact are zeroed. Finally, the signal is reconstructed from the corrected set of coefficients. The advantages of wavelet filtering are in its simplicity and performance: it eliminates the need for the detection of MAs, it can be fully automated, and it significantly reduces spikes and drift noise [4].

In this study, we present a new combined artifact correction approach, wavelet and CBSI correction (WCBSI). We recorded extensive evoked fNIRS data from the motor cortex while participants performed a tapping task, and distorted the measurements with head movements of different extents. In parallel, we measured head movements with accelerometers. The design of the experiment allowed us to empirically estimate the “ground truth” hemodynamic response for HbO and HbR. We used this “ground truth” as a reference signal to evaluate the performance of the proposed WCBSI and several other popular MA corrections on four quality metrics and two levels of head movements. We compare WCBSI performance to seven established MA correction approaches and demonstrate that it has reliably superior performance in the cases examined here. The dataset is available as a reference for future developments and comparisons of MA correction techniques (https://www.doi.org/10.17605/OSF.IO/3A9Q6 (accessed on 24 March 2023)).

## 2. Materials and Methods

### 2.1. Motion Artifact Correction Methods

All MA correction methods used in this study are integrated into the HOMER3 toolbox. For brevity, we will only cover the CBSI and wavelet methods in more detail as they are at the core of our improved MA correction approach. All other methods are only briefly outlined, and we refer the reader to publications with more detailed explanations. Figure 1 depicts the data flow through the processing steps required for each different MA correction approach [26]. Table 1 lists the user selectable parameters of each algorithm. Their total number adds up when multiple algorithmic steps are combined, for example when MA detection and correction are concatenated in spline, splineSG, and tPCA.

PCA takes the covariance between N sensors and generates N uncorrelated (orthogonal) components. The components are sorted by the amount of variance they account for in the original data. Because MAs typically have larger amplitudes than brain signals, they are expected to be captured in the larger components. MA correction is performed by finding the first M components that represent MA variance and removing them from the signal. A user-defined threshold specifies the minimal amount of variance MA components explain. If the MAs are statistically independent of brain activation, they may end up in separate components. However, this is not guaranteed. A critical factor that determines PCA’s performance is the number of measurements available to estimate the covariances, and the number of components to be removed [18].

TPCA is designed to prevent over-correction, which occurs for example when MA and brain signals do not separate well in PCA [19]. In order to obtain a better representation of Mas, the method confines PCA estimation to previously detected MAs. Therefore, MA detection has to be performed before the PCA is applied and the movement intervals are corrected. This method has five user parameters: four from the motion detection and one from PCA.

In HOMER3, MARA is implemented in the correctSpline function, and we will denote it as the function in the rest of the manuscript [14]. The algorithm initially finds intervals with artifacts in each channel, fits a cubic spline to it, and subtracts the fitted spline function to remove the artifact. Effectively, it has five user parameters: four for the motion detection and one to control the smoothness of the spline (one for a cubic spline and zero for a linear fit).

Savitzky–Golay filtering [22,24] is implemented in the correctSplineSG function in HOMER3, and we will denote it like this in the rest of the manuscript. As the name indicates, the function combines the spline-based MARA algorithm [14] with Savitzky–Golay filtering in a second step. The SG filtering step obtains a least squares fit of a third-order polynomial to a data interval of a pre-specified length. It replaces the original value at the center of the interval by the value of the fitted functions. This process is repeated until all data points are smoothed. Effectively, this algorithm has six user parameters: four for motion detection, one for spline interpolation, and one, the length of the fitting interval, for the SG filter.

RLOESS [23,24], as implemented in HOMER3, uses the MATLAB function ‘smooth’. In the first step, this function fits the second-order polynomial to an interval of a pre-specified length of the data, and replaces the original data with the smoothed data, similar to SG filtering. However, one important difference is that the data pointing further away from the current interval center are down weighted as a function of their distance. In a second step, the residuals between the smoothed and the original data are calculated, and the fitting procedure is repeated but now with an additional weight that down weights data points that were not well fitted in the first step. Data points that are more than six standard deviations from the fitted value are excluded as outliers, i.e., they receive the weight zero. This makes the fit robust against outliers. The two steps can be iteratively repeated to increase the smoothness of the fit. This algorithm has one user parameter: the length of the data interval used for fitting. All other parameters are fixed to standard values.

The wavelet transform (WT) is well suited to separate temporally localized signals with different dynamics. It achieves that by tiling the time-frequency plane in a way that optimizes the trade-off between frequency and time resolution, i.e., wavelets sensitive to rapid signal variations are typically shorter than wavelets sensitive to slow variations. This is in contrast to short time Fourier transform, which has constant temporal resolution over all frequencies due to the fixed analysis interval for all sinusoids. Another advantage of WT is that it allows for different basis functions, which can be chosen to approximate the spiky shape of Mas. With an appropriately chosen wavelet, the WT is sensitive to the shape and the dynamics of Mas, which helps to separate them from the brain-related fNIRS signal [25]. The HOMER3 wavelet-based motion correction uses the computationally efficient discrete wavelet transform with the db2 wavelet, which has a spiky shape. The function applies discrete WT to decompose the fNIRS signal of each channel at multiple scales. It then calculates the distribution of the obtained wavelet coefficients, and zeroes coefficients with extreme values (exceeding a pre-specified inter-quantile range) as they are assumed to reflect artifacts. Finally, the correct time series of each channel is reconstructed by applying the inverse discrete WT to the wavelet coefficients. The method has one user parameter: the width of the inter-quantile range, which specifies the width of the central range of the wavelet coefficients that are retained. All other parameters are fixed to standard values.

The HbO and HbR signals induced by brain activity are negatively correlated. Neuronal activity triggers the increase in arterial, HbO-rich blood supply via neurovascular coupling. As a consequence, the blood volume increases locally and the relative HbO concentration increases while the relative HbR concentration decreases. Conversely, motion changes the sensor signal for both IR wavelengths in a similar way. Therefore, motion artifacts are characterized by positively correlated HbO and HbR signal changes. This difference is the basis of the correlation-based signal improvement (CBSI) [20] implemented in HOMER3. The algorithm requires additional assumptions for signal correction. In order to make them transparent, we briefly review the approach implemented in HOMER3 based on [20]. The correction assumes that the observed HbO and HbR signal time series *X* and *Y*, respectively, are composed of contributions from three sources: (1) the true signals *X*_0_ and *Y*_0_ generated by HbO and HbR concentration changes in the brain, (2) movement-related noise causing similar effects *F* on HbO and HbR but with different scaling captured by a positive constant factor *α*, and (3) instrument noise. The observed HbO and HbR measurements *X* and *Y* can then be written as
(1)$$X=X_{0}+\alpha F+Noise,\;and\;Y=Y_{0}+F+Noise$$

It is further assumed that the movement-related noise *F* and the instrument noise are independent of each other and the brain signal. The correction algorithm focuses on the movement-related noise *F* and assumes that the instrument noise was removed by some other approach such as filtering. This is why we suggest to apply wavelet denoising before CBSI. With the assumption that the true *X*_0_ and *Y*_0_ are perfectly negatively correlated, we can express one as a negatively scaled version of the other *X*_0_ = −*βY*_0_ with *β* as the scaling factor. Based on empirical observations, the authors of [20] argue that the scaling *α* of the movement-induced noise is similar to the scaling *β* of the true HbO/HbR signals, i.e., *α = β*. With these assumptions, they derive the following equations:(2)$$X_{0}=\frac{1}{2}\left(X-\alpha Y\right),\;and\;Y_{0}=\frac{-1}{\alpha }X_{0}$$
for the estimation of the “true” motion corrected HbO/HbR signals observed. Importantly, setting *α = β* makes it possible to estimate α as the ratio between the standard deviation of the observed HbO and HbR signals, i.e., *α = std(X)/std(Y)*. The method has no user parameters.

The different MA corrections methods have a different number of free user parameters which allow for adjustments. The parameter settings used in this study are listed in Table 1. We chose typical standard parameters, and the same parameters were used for processing steps implemented in different MA correction pipelines.

In this study, we suggest to combine the wavelet-based correction with CBSI to further improve artifact correction. In the combined wavelet CBSI (WCBSI) correction, we apply wavelet correction first. The reason is that it reduces, among other noises, instrument noise that would violate the assumptions of CBSI.

### 2.2. Experimental Procedure and fNIRS Data Recording

The general aim of the experimental procedure was to obtain data that (a) reflect the situation in an experiment at (b) different levels of artifact severity, but at the same time (c) allow for the estimation of a “ground truth” reference signal. Moreover, we wanted to collect (d) a larger than usual amount of data in order to obtain (e) good estimates for the MA correction performances at the single participant and at the group level with (f) fNIRS-relevant variations across participants. Finally, we wanted to obtain (g) an estimate of the relative strength of the head movements actually performed. Therefore, we recorded an accelerometer attached to the participants’ heads.

#### 2.2.1. Participants

Twenty healthy young adults (aged 22–37, thirteen females) participated in this study. Nine had black hair, ten had variations of brown hair, and one had red hair. In addition, their hair density differed from dense to medium. The participants had different ethnicities (Asia, Africa, and Europe). All participants provided written informed consent. The IRB of the Carl von Ossietzky Universität Oldenburg approved the experimental protocol under the code Drs. EK/2020/021.

#### 2.2.2. Experimental Setup and Procedure

Participants were seated in front of a 24″ DELL (LCD) monitor (1920 × 1200 px resolution and reduced brightness) at a viewing distance of approximately 70–80 cm in a dimly lit and acoustically isolated room (see Figure 2b). The participants were asked to put their hands on the table during the experiment. They were instructed to begin tapping with both hands on the table, with approximately one to two taps per second, when the word “tapping” appeared on the screen.

The experiment started and ended with a 120 s long baseline measurement (Figure 3). Then, hand tapping was performed during 10 s long intervals followed by a rest interval (20 s) to allow the slow hemodynamic response to settle to baseline between consecutive tapping intervals. The fNIRS optodes were positioned according to the international transcranial optode positioning system (10/10 and 10/20) [27], with 3 cm source–detector distance. The detectors were on positions C3/C4, and the emitters on positions C1/C2, C5/C6, FC3/FC4, and CP3/CP4, as shown in Figure 2a. This way they covered the participants’ motor cortex. During the first tapping interval in a sequence, participants kept their head steady, which we denote as being tapping “no head movement” (NHM). During the second tapping interval, they performed small head movements (SHM, right-left flexion, and head rotations with steady shoulders), and during the third interval they performed large head movements (LHM), with movements of similar directions but larger amplitude. This sequence was repeated 25 times resulting in 25 repetitions of each experimental condition. The whole experiment took approximately 41.5 min. We made sure that neither optode positions changed nor that the signal quality deteriorated during the experiment. Each participant practiced the tasks for 5–10 min before starting the actual recordings. The experimental procedure was controlled with Psychtoolbox 3 [28].

#### 2.2.3. fNIRS and IMU Data Recording and Processing

We used a portable Artinis Medical OctaMon 8-channel system. This system is designed to measure fNIRS under free field conditions. It has eight light sources with standard wavelengths of 760 and 850 nm and two receivers, forming eight optodes in total. Each optode was sampled at 10 Hz. We performed multiple quality checks before the recordings began. First, we checked via the recording software (Oxysoft v3.0.103.3) that the light intensity was in the recommended range. Then, we assessed potential contributions from environmental light by measuring the amount of light in the detectors when the device’s light sources were turned off. This contribution amounted to less than 1% of the total signal. Finally, we confirmed that the wavelengths of each source were at 760 ± 5 nm and 850 ± 6 nm.

Furthermore, we used an MPU-6050 motion tracking sensor with a 3-axis accelerometer to record the movement during each task by fixing it between CZ and FZ, as shown in Figure 1a. We sampled the MPU-6050 sensor signals with an Arduino Uno board at 10 Hz, the same sampling frequency as the fNIRS signal, and sent it via USB to the fNIRS recording PC.

#### 2.2.4. Data Processing and Movement Correction

The data processing was conducted with the HOMER3 MATLAB toolbox [29], with MATLAB R2020a. The block diagram in Figure 1 depicts the sequence of processing steps for all correction techniques applied in this study.

In the first step, we cut the optical density (OD) recordings into epochs around the tapping signal shown on the screen (2 s before to 20 s after). Next, we performed an initial head motion artifact detection step on the raw OD recordings for spline, splineSG, and tPCA. We then applied seven motion correction techniques (PCA, tPCA, spline, spline SG, RLOESS, wavelet, CBSI). Note that PCA, tPCA, spline, spline SG, RLOESS, and wavelet corrections are performed on the optical density data. The clean epochs were then converted to HbO/HBR concentration changes using the modified Beer–Lambert law. The manufacturer’s software (Oxysoft v3.0.103.3) implements the formula suggested by Scholkmann and Wolf [30] for conversion that uses an age-dependent differential path length factor. After entering the participant’ ages, this factor amounted on average to 6.28 ± 0.5 for the wavelength 760 nm and 5.85 ± 0.3 for the wavelength 850 nm. CBSI assumes a negative correlation between HBO and HbR. Therefore, it is necessary to apply the modified Beer–Lambert law before movement correction. Then, the converted epochs were band-pass filtered and averaged within the movement conditions (NHM, SHM, and LHM) [26,31]. All parameters of all algorithms and processing steps are summarized in Table 1.

## 3. Quality Metrics for Comparison among MA Correction Algorithms

In order to quantify the performance of the different motion correction algorithms, we calculated four metrics on the averaged HbO and HbR signals that compared the “ground truth” reference signal, recorded without head movements (NHM), to the signals recorded with head movements (SHM and LHM) after MA correction. We used participant averaged epochs for quality metric calculation because this mimics the standard situation in event-related experiments where epochs are averaged over repetitions of an experimental condition. The MA correction was applied on the single trial data.

(1)Pearson’s correlation coefficient R was calculated between the averaged *HRF* of the reference signal HRF (NHM) and $\hat{HRF}$ of the movement-contaminated signals (SHM and LHM). Pearson correlation measures the similarity of the shapes of two signals and is scaled between −1 and 1.

(2)Rooted Mean Square Error (*RMSE*) measures the unscaled average deviation between two signal time series. It was calculated with the following equation:
(3)$$RMSE=\sqrt{\frac{\sum_{i=1}^{N}{\left(HRF\left(i\right)-\hat{HRF}\left(i\right)\right)}^{2}}{N}}$$
Here, *N* is the number of samples in an epoch and *i* is the sample count.

(3)Mean Absolute Percentage Error (*MAPE*) measures the deviation in relation to the momentary strength of the reference signal. It was obtained with the formula:


(4)
$$MAPE=\frac{1}{N}\sum_{i=1}^{N}\left|\frac{HRF\left(i\right)-\hat{HRF}(i)}{HRF\left(i\right)}\right|$$


(4)The area under the curve difference (Δ*AUC*) is a global measure that compares the overall deviation from the baseline of two curves. It was obtained with the formula:


(5)
$$\Delta AUC={AUC}_{HRF}-{AUC}_{\hat{HRF}}$$


We report for each quality measure the mean across participants, separately for HbO/HbR and the movement condition. In order to derive summary statistics across MA algorithms, we rank the mean MA correction performances on each measure separately. We switch to ranks because they provide a robust new performance measure (e.g., against non-linearities) that can be combined over quality measures that have different scales and numerical ranges. One summary statistic we report is the mean rank of each MA correction algorithm across quality measures, signal type, and movement condition, plus the corresponding standard deviation. This provides a basic global performance statistic. Other summary statistics include rank probabilities and a statistical significance test for the observed rankings. Therefore, we fitted a Plackett–Luce model (PLM) to the 16 rankings (2 fNIRS signals * 2 movement levels * 4 quality measures) obtained for each MA correction algorithm using the R-package “Plackett–Luce” [32]. The Plackett–Luce framework models the observed ranking as a sequence of choices among alternatives (the MA algorithms) with different “worth”. Algorithms with higher worths can be expected to be better ranked. The package provides probabilities for the first rank for each algorithm, as well as standard errors and statistical significance of the worth of each algorithm.

## 4. Movement Analysis

Newton’s law linearly relates acceleration measured by the IMU to the force that displaces the optodes on the head. We quantified the amount of acceleration in each epoch by taking the geometric mean over the three accelerometer axes for each sample and then averaged this over all samples using the equation:(6)$$M=\frac{\sum_{i=1}^{N}\sqrt{X_{i}^{2}+Y_{i}^{2}+Z_{i}^{2}}}{n}$$

This is the mean scalar acceleration per epoch. Here, *i* is the number of samples in an epoch.

## 5. Results

### 5.1. Movement Analysis

The accelerometer data provide an estimate of the forces acting on the optodes in the different movement conditions. Larger forces can displace optodes on the scalp by a larger amount and thereby create larger MAs. In this acceleration analysis, we first calculated the mean acceleration per epoch, averaged them within the conditions (NHM, SHM, LHM), and then across participants. The results are shown in Figure 4. As expected, the smallest acceleration was measured when the head was held steady (NHM) and the highest with largest instructed movements (LHM). Intermediate movements (SHM) produced intermediate accelerations.

### 5.2. fNIRS Movement Artifacts

The instructed head movements introduced various types of MAs [26]. Figure 5a shows, as an illustrative example, the time course of the raw HbO and HbR signals measured in one optode of one participant for the duration of a full experiment. The HbO signal increases due to tapping are clearly visible as an oscillation with constant frequency over the whole experiment, indicating the good signal-to-noise ratio of the recording. The large spikes occurring approximately every third period reflect MAs caused by LHM. They can have qualitatively different temporal fine structure such as spikes, signal drifts, or signal shifts. Examples are shown as small insets. While the movement artifacts shown in Figure 5a can be detrimental in time-resolved BCI-type analyses [12], they may even deteriorate the signal after extensive averaging. Figure 5b shows the time courses of HbO and HbR for all three conditions (NHM, SHM, and LHM). Each curve was calculated by averaging over 25 movement epochs. The mean over the NHM epochs exhibits the expected delayed responses with a single peak but high frequency ripples. The latter likely reflect residual instrument and physiological noise. The LHM average clearly has residual spikes that severely distort both the HbO and HBR averages. Shape distortions in the HbO and HbR signals are even more obvious in the SHM averages.

### 5.3. Comparison of Movement Correction Methods

We measured the quality of the different movement correction algorithms with respect to a reference signal; the average of the NHM epochs represents the most “typical” empirical brain signal expected in our experiment. Note that residual uncorrelated noise should be largely averaged out in these repetitions. An ideal motion correction would result in a response curve that is very similar to the reference signal. We computed four different quality measures (R, RMS, *MAPE*, and Δ*AUC*; see methods), each capturing different types of ‘similarity’ between the reference signal and the average movement corrected signal from the two movement conditions SHM and LHM. The quality measures were computed individually for each participant, and we report the average for the participants here. We expressed noise reduction in amplitude decibels with the values of the measure obtained with averaged uncorrected NHM data as a reference.

The first measure is Pearson correlation (R), which captures the overall similarity of the shapes of the reference and the (corrected) head motion contaminated signals. R is a normalized value where one indicates that the shapes of two curves are perfect reproductions of each other, whereas zero means they are unrelated. Note that R does not capture amplitude differences. Figure 6 shows the correlation coefficients for the different correction approaches for HbO and HbR separately. Panel (a) is for SHM and (b) is for LHM. The highest correlations are obtained with the suggested WCBSI approach and, importantly, the correlations are similar between the two levels of head movement (SHM: R = 0.85 for HbO and HbR; LHM: R = 0.82 for HbO and HbR). These results suggest that the WCBSI approach is remarkably robust over levels of head movements, and is particularly efficient for the HbR signal.

Except for CBSI and WCBSI, the corrected HbR signals have lower correlations with the reference signal than the HbO signals after correction. This suggests that most correction algorithms struggle to recover the shape of the HbR response. Moreover, the correction of most other algorithms (including CBSI) is worse for the larger head movements, at least for the HbR signal, and the obtained correlations remain at a similar level as the uncorrected HbO signal.

Rooted mean square error (*RMSE*) measures the unscaled average absolute deviation between the reference signals and the (corrected) signals contaminated with head movements. Note that smaller *RMSE* scores indicate better movement artifact correction. Figure 7 shows the *RMSE* for the different correction approaches for HbO and HbR separately. Panel (a) is for SHM and (b) is for LHM. Similar to Pearson correlation, the highest errors are observed with the uncorrected signals and the best results (lowest error) with the WCBSI method (SHM: 1.8 × 10^−5^- for HbO and 1.2 × 10^−5^ for HbR; LHM: 2.4 × 10^−5^ for HbO and 1.5 × 10^−5^ for HbR). In decibels, this corresponds to a noise reduction of −12.2 dB for HbO and −10.6 dB for HbR with SHM and with LHM −15.1 dB for HbO and −13.9 dB for HbR. For larger head movements, the *RMSE* is comparable between RLOESS and WCBSI, but for smaller movements there is still a clear improvement in WCBSI over RLOESS (HbO: 53%, HbR: 17% improvement). Note that the generally lower *RMSE* for the HbR signals is due to the generally lower signal variation compared to HbO.

Mean absolute percentage error (*MAPE*) expresses the correction error as a percentage of the current reference signal strength. Consequently, smaller numbers are better. Figure 8 shows the *MAPE*s for the different correction approaches for HbO and HbR separately. Panel (a) is for SHM and (b) is for LHM. Note that for constant noise amplitudes, *MAPE* will increase when the amplitude of the reference signal decreases (e.g., close to the in the baseline signal). Again, WCBSI produces favorable results over the range of head movements (SHM: 24% for HbO and 23% for HbR; LHM: 53% for HbO and HbR). Again, the performance is very similar for HbO and HbR. This corresponds to a noise reduction of −15.7 dB for HbO and −8.2 dB with SHM and with LHM −10 dB for HbO and −6.3 dB. Only splineSG in the LHM condition produces a slightly better result for the HbR signal with 37% *MAPE*. However, for HbO, splineSG correction is worse than WCBSI for both SHM and LHM (59% and 74%).

Area under the curve difference (Δ*AUC*) measures the difference between the integrals of the reference signals and the (corrected) signals contaminated with head movements. This is a global measure that can, for example, detect signal rescaling. Reported alone, it can be hard to interpret. Figure 9 shows the Δ*AUC*s for the different correction approaches for HbO and HbR separately. Panel (a) is for SHM and (b) is for LHM. Note that the HbR signals were multiplied by −1 to compensate for the sign flip. As with the other measures, WCBSI produces favorable results over the range of head movements on Δ*AUC* (SHM: 3.3 × 10^−4^ for HbO and 1.1 × 10^−4^ for HbR; LHM: 1.0 × 10^−4^ for HbO and 1.4 × 10^−4^ for HbR). In decibels, this corresponds to a noise reduction of −9.6 dB for HbO and −14.9 dB with SHM and with LHM −21.4 dB for HbO and −13.7 dB for HbR. Note that for HbR, the Δ*AUC* performance is very similar between movement conditions. For the combination SHM and HbO, WCBSI is still in the middle field after wavelet, CBSI, and RLOESS. For all other combinations of movement and fNIRS signal, WCBSI provides the best Δ*AUC* measures.

In order to summarize the results in a way that generates recommendations for users of MA correction algorithms, we determined the MA correction rank for each algorithm in each combination of signal type (two levels), movement (two levels), and evaluation score (four levels). This resulted in 16 rank scores for each algorithm. We assume that most researchers will be interested in an algorithm that exhibits good and reliable MA correction. The first criterion can be assessed by sorting the algorithms according to their mean rank over all scores (conditions). The results are shown in Table 2, with rows sorted by the second column. WCBSI, the algorithm proposed here, clearly has the best mean rank. It is followed by RLOESS and CBSI by some margin. The two worst approaches to deal with MAs are “uncorrected” (doing nothing) and PCA. The second criterion, reliability, can be measured by taking the standard deviation across rank scores. An algorithm that produces a certain performance level consistently over the conditions will exhibit low rank standard deviation. Table 3 again shows mean ranks and rank standard deviation, but the rows are now sorted by rank standard deviation (column 3). Again, WCBSI exhibits the smallest standard deviation, indicating that it provides most consistent results over all conditions. For the remaining methods, an approximate pattern emerges where the better the mean rank, the higher the standard deviation. This could indicate that the better algorithms perform particularly well in some, but badly in other conditions. For example, RLOESS produces very good *RMSE* scores (Figure 7), which is the criterion it optimizes. However, the shape of the resulting signal may deviate considerably from the reference shape, as indicated by the algorithm’s low correlation scores in Figure 6. Interestingly, the second most reliable approach is ‘uncorrected’, where nothing is done to the signal. This approach produces consistently bad results.

In sum, the analysis of mean rank and rank standard deviation suggests the WCBSI algorithm as the most favorable MA correction algorithm that can reliably produce excellent results over different MA levels, for the HbO and the HbR-signal, and for different correction quality measures. Not treating MAs consistently produces the worst results.

The Plackett–Luce modeling of rank data allows for the derivation of several statistics of rankings [32]. In Figure 10, we plot the worths of the different MA correction algorithms with reference to “uncorrected”, for which the mean worth was set to zero. Higher worths indicate better performance rank. WCBSI has the highest worth and “uncorrected” the lowest. The worth of all other algorithms is at an intermediate level. WCBSI clearly has the highest probability of 78.8% to obtain the first rank in the MA correction quality ranking. In contrast, the other algorithms have a probability between 0.2% and 3% to be found in the first rank. Moreover, WCBSI was the only MA correction with a significantly higher rank than the mean rank (*p* < 0.0001). From these results, we again conclude that WCBSI produces systematically better results on the four quality measures than the seven other algorithms tested here.

## 6. Discussion

The central aim of our study was to assess and compare the performance of the newly proposed WCBSI MA correction approach to other popular approaches, with a realistic amount of fNIRS data that were recorded in a realistic experiment. In sum, over all evaluation measures tested, WCBSI provided the best and most reliable MA correction performance over all levels of head movement strengths for both fNIRS signal types (HbO/HbR). WCBSI provided the highest correlations in combination with the lowest *RMSE*. This suggests that it is highly capable of recovering not only the shape but also the amplitude of the brain responses in both the HbO and the HbR signals. If no specific preference for a particular MA correction type can be a priori specified, our results suggest that WCBSI should be used to obtain consistently good correction results over magnitudes of movement and signal types. Notably, the WCBSI method consistently produced the best correction results for the HbR signal. However, our analysis also showed that the worst thing to do with MA contaminated recordings is to leave the data uncorrected. This holds even when a considerable number of trials can be averaged.

The suggested WCBSI approach for MA correction has several advantages over previous approaches. One advantage is the good performance with different levels of MA artifacts, evaluation measures, and, importantly, with HbO and HbR signals. Moreover, the WCBSI MA correction was performed on the HbR and HbO signals comparably. This indicates that the assumption that HbO and HbR are negatively correlated holds for our data. This is important because the other algorithms tested here had problems with recovering the shape of the HbR signal (Figure 6). Another advantage is its simplicity. It has only one user parameter, the wavelet threshold. This parameter can be estimated from the data, or the standard parameter of HOMER3 (1.5) can be used. The performance of the wavelet algorithm appears to be relatively robust against its variations [33]. Therefore, the proposed WCBSI algorithm has the potential to be automated. Yet another important advantage is that implementations of the wavelet [25] and CBSI [20] algorithms are already available in software, for example in HOMER3.

The runtime of WCBSI is mostly determined by the wavelet part of the combined algorithm (Appendix A). With 65 s for more than 40 min of recordings in 8 channels (8 HbO and 8 HbR signals), it is not the fastest approach, but it is in a similar order of magnitude as most other algorithms (range 16–690 s) and is considerably faster than RLOESS which required 690 s on the same data. Since the algorithm is applied separately on channels, the runtime scales linearly with their number. Moreover, the separate processing of channels allows parallelization for further acceleration of processing. These considerations suggest that WCBSI will have an acceptable runtime on data from a typical whole head fNIRS system, with about 10 times as many channels than our study used.

MA correction algorithms are frequently evaluated on synthetic data [34,35,36,37] or hybrid data with a synthetic ground truth signal [14,17,20,24,33]. This is, on the one hand, understandable because not all methods developers may have access to fNIRS recordings. Moreover, knowing the ground truth of the to-be-recovered signal has advantages for the evaluation of the performance of the MA correction algorithm. On the other hand, using synthetic or semi-synthetic data has limitations. First, the signal statistics may differ from signals recorded in real experiments. For example, artifacts other than MAs (e.g., heartbeat and other physiological artifacts) will be likely missing and the variability of brain responses is underestimated. This could lead to biased performance estimates. Second, fNIRS measures two types of signals (HbO and HbR), but the simulations may not reproduce both signal types equally well and enforce theoretical assumptions that require empirical support during method evaluation (e.g., negative correlation between HbO and HbR). This may not only introduce a bias towards one type of signal (often HbO) but may reduce the usefulness of the evaluation of MA correction methods combining HbO and HbR, such as the CBSI method which is part of our proposed WCBSI approach. Moreover, studies using recorded brain data tend to use rather small datasets with only few participants and/or very little data per participant [25,33,38]. This bears the risk of biased results that are hard to reproduce [39]. Our dataset attempts to avoid such pitfalls by including a relatively MA-free condition without instructed head movements, plus two additional levels of head movements. The experimental task is simple, and each condition is repeated 25 times leading to 41 min of data per participant. These data are enough to estimate individual ‘ground truth’ average hemodynamic brain responses that can be used as a reference signal for MA correction algorithm comparison. With 20 participants, the dataset should be large enough to discover relevant performance differences among MA correction algorithms. In addition, the dataset will include IMU recordings from the head, synchronized with the fNIRS recordings. We think that our dataset could be helpful for other researchers as a reference dataset to evaluate and develop MA correction algorithms with other measures and approaches implemented here. Therefore, we will make it available to the research community (https://www.doi.org/10.17605/OSF.IO/3A9Q6 (accessed on 24 March 2023)). We encourage other researchers to release rich fNIRS datasets with different tasks to make a variety of fNIRS measurement scenarios available. This will help to improve the comparison of MA correction algorithms in different types of experimental settings.

When comparing MA correction algorithms over multiple experimental conditions and quality measures, it is important to span a range of scenarios. The downside of this approach is that it can produce complex and hard to interpret results because the ranking among MA algorithms can strongly depend on the particular variant of data and quality measure, e.g., comparing Figure 6, Figure 7, Figure 8, Figure 9, [4,17,24]. In order to reduce this complexity, we presented for each algorithm a rank order statistic that was calculated over the ranks and obtained in every single combination of fNIRS signal/head movements and level/quality measures that were tested. This is an easily interpretable and non-parametric summary of the results. We find that the proposed WCBSI algorithm performs favorably in many combinations. The mean ranking we obtained is in broad correspondence with previous studies. For example, RLOESS, CBSI, wavelet, and splineSG are often reported among the more favorable algorithms, whereas spline and PCA as less favorable and dependent on the data [4,17,24,35]. Importantly, our analysis is, to our knowledge, the first to explicitly quantify the reliability of the tested MA correction algorithms by introducing rank standard deviation. This measure is independent of mean rank, easy to interpret as reliability of performance, and provides important insights. First, the new WCBSI has best mean rank and also has low rank standard deviation, meaning it performs reliably well. Another approach with low rank standard deviation is to leave the data uncorrected. This approach produces reliably bad results. For the other established approaches tested here, our results suggest an approximate negative relationship between mean rank and rank standard deviation. The better the mean rank of an algorithm, the less reliable its level of performance is when tested over different movement levels, quality measures, or fNIRS data types. This variability in quality measures may also explain why most MA correction algorithms had similar worths in the Plackett–Luce model; good results on one measure may average out with less favorable results on another. The proposed WCBSI is the notable exception of this pattern with a favorable mean rank and a low rank standard deviation.

## 7. Conclusions

In this study, we proposed WCBSI as a new approach to motion correction in fNIRS signals, which combines two existing MA correction methods to handle many types and magnitudes of motion artifacts on four commonly used quality metrics (R, *MAPE*, *RMSE*, and Δ*AUC*) with consistently favorable performance. When compared to a range of popular MA correction techniques, WCBSI outperformed them in the quality and the reliability of the MA correction.

## Figures and Tables

**Figure 1 sensors-23-03979-f001:**
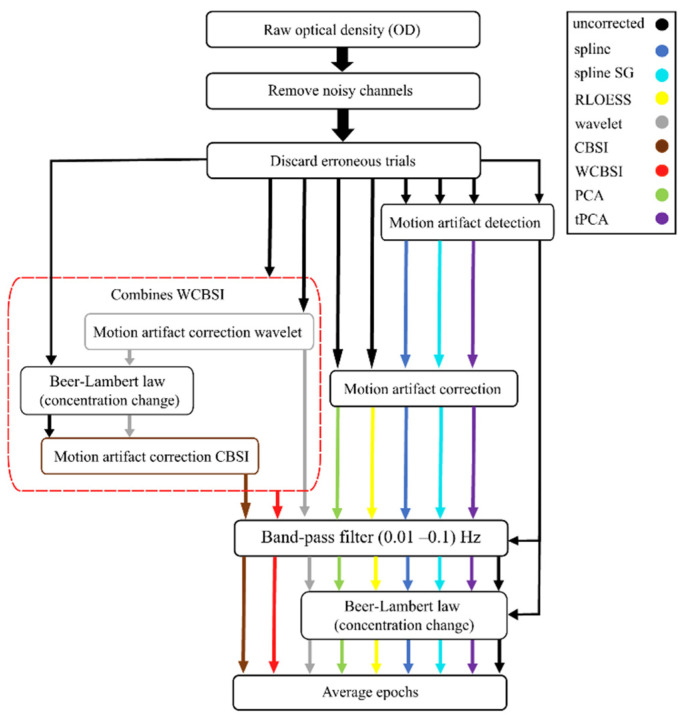
The flow of data for all MA correction techniques tested here. The rounded rectangles indicate the processing steps applied to the data. The data flow of each technique is indicated by color-coded arrows: black for uncorrected data, blue for spline, cyan for spline SG, yellow for RLOESS, gray for wavelet, brown for CBSI, green for PCA, purple for tPCA, and red for WCBSI correction. The red dashed box indicates how processes are combined in the WCBSI correction. All parameters used in the respective MA correction algorithms are listed in Table 1.

**Figure 2 sensors-23-03979-f002:**
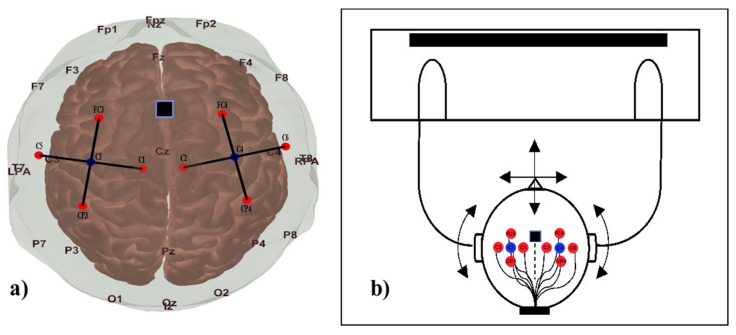
(**a**) Optode placement. We recorded using eight optodes placed according to the 10/20 system in positions approximately located over the hand area of the primary motor cortices. The two detectors placed at C3 and C4 are marked in blue, and the sources located at FC3, FC4, CP3, CP4, C1, C2, C5, and C5 are marked in red. The black lines approximate the channels. The accelerometer was located in the FCz position and is indicated by a black box. (**b**) The participant is seated in front of the monitor with his/her hands on the table. Arrows indicate the approximate directions of the head movements.

**Figure 3 sensors-23-03979-f003:**
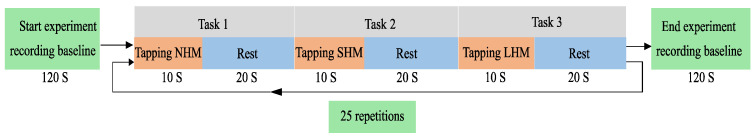
The experimental procedure and sequence of the tasks. The participants first performed hand tapping without head movements for 10 s (NHM), then, after a 20 s rest period, hand tapping with small head movements (SHM) for 10 s, and, after a 20 s rest period, hand tapping with large head movements (LHM) for 10 s, followed by a 20 s rest period. This sequence was repeated 25 times. A baseline was recorded at the beginning and the end of the experiment.

**Figure 4 sensors-23-03979-f004:**
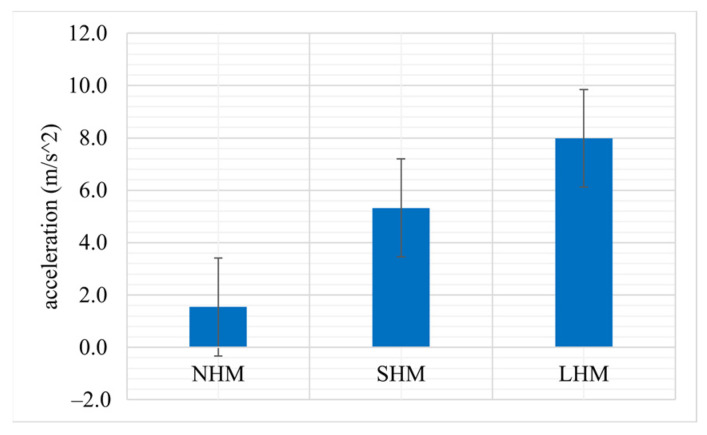
Mean head acceleration over participants for all three movement conditions (NHM, SHM, LHM). Error bars indicate standard error across participants.

**Figure 5 sensors-23-03979-f005:**
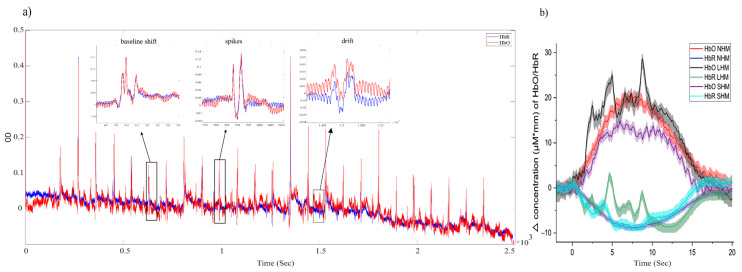
(**a**) An example raw time course of optical density signals of one participant (red: HbO, blue: HbR). The small insets exemplify different types of MA (baseline shifts, spikes with high amplitude and fast dynamics, and slow drifts with lower amplitude and slower dynamics) at a finer temporal scale. (**b**) Corresponding average uncorrected epochs of the raw concentration change of HbO/HbR for the three movement conditions. Different levels of movement artifacts distort the tapping related response to different extents, even after averaging. Colors codes conditions: red for HbO in NHM epochs, blue for HbR in NHM, black for HbO in LHM epochs, green for HbR in LHM, purple for HbO in SHM, and cyan for HbR in SHM. Error bands indicate standard error across trials.

**Figure 6 sensors-23-03979-f006:**
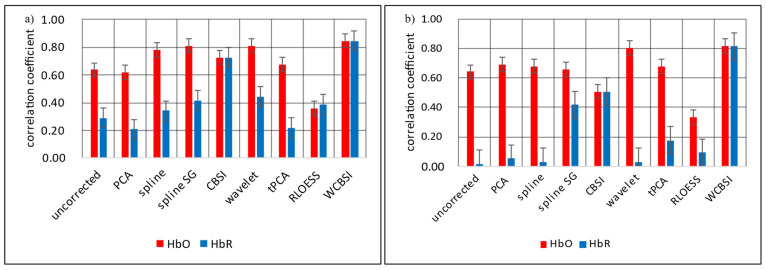
Average Pearson correlations of the MA corrected HbO and HbR signals with the reference signals. These are shown in (**a**) for the SHM responses and in (**b**) for the LHM responses. The correlations were calculated individually for each participant and then averaged. Correction types are uncorrected, principal component analysis (PCA), spline interpolation, spline interpolation plus Savitzky–Golay filtering (splineSG), correlation-based signal improvement (CBSI), wavelet transform, targeted principal component analysis (tPCA), robust locally estimated scatter plot smoothing (RLOESS), and our new combination of wavelet and CBSI correction (WCBSI). Error bars indicate standard error across participants.

**Figure 7 sensors-23-03979-f007:**
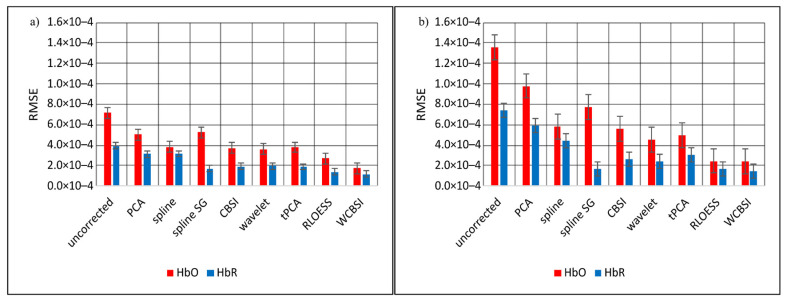
Average rooted mean square error (*RMSE*) between the MA corrected HbO and HbR signals and their respective reference signal. This is shown in (**a**) for the SHM responses and in (**b**) for the LHM responses. The *RMSE*s were calculated individually for each participant and then averaged. Error bars indicate standard error across participants.

**Figure 8 sensors-23-03979-f008:**
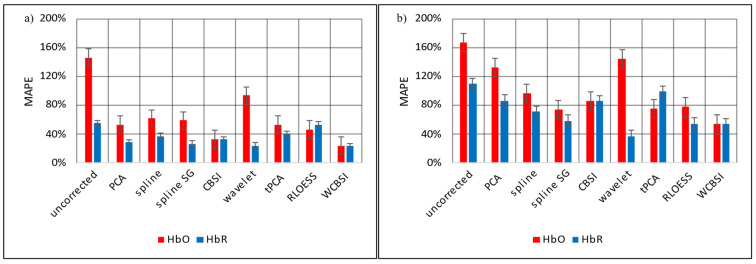
Mean absolute percentage error (*MAPE*) between the MA corrected HbO and HbR signals and their respective reference signals. This is shown in (**a**) for the SHM responses and in (**b**) for the LHM responses. The *MAPE*s were calculated individually for each participant and then averaged. Error bars indicate standard error across participants.

**Figure 9 sensors-23-03979-f009:**
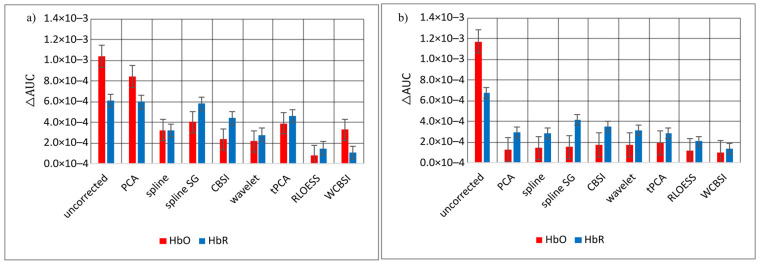
Mean difference between areas under curve (Δ*AUC*) of the MA corrected HbO and HbR signals and their respective reference signals. This is shown in (**a**) for the SHM responses and in (**b**) for the LHM responses. The Δ*AUC*s were calculated individually for each participant and then averaged. Error bars indicate standard error across participants.

**Figure 10 sensors-23-03979-f010:**
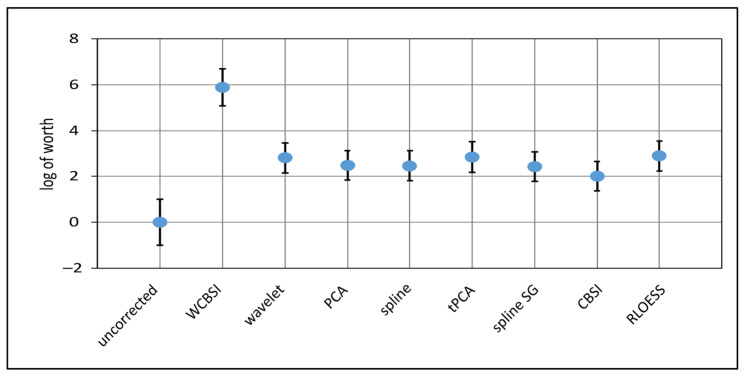
The log worth of the different MA correction methods obtained by fitting a Plackett–Luce model to the ranking data. Higher values indicate better performance of the correction algorithm. WCBSI has the highest worth and “uncorrected” the lowest. All other correction algorithms have similar worths varying around the mean. All worths are referenced to “uncorrected”, for which the mean was set to zero. Error bars indicate standard errors of the mean.

**Table 1 sensors-23-03979-t001:** User parameter settings for every HOMER3 function used in the processing stream.

Name	Function	Parameters and Values
Channel rejection	hmrR_PruneChannels	dRange (1 × 10^−4^–1 × 10^7^), SNRthresh = 1, Sdrange = (0.0–45.0)
Motion detection	HmrMotionArtifactByChannel	tMotion = 0.5 Sec, tMask = 1.0 Sec, SDEVThresh = 20, AMPthresh = (0.05–0.5)
PCA	hmrR_PCAFilter	nSV = (0.96 ± 0.02)
tPCA	hmrR_MotionCorrectPCArecurse	tMotion = 0.5 Sec, tMask = 1.0 Sec, SDEVThresh = 20, AMPthresh = (0.1–0.5), nSV = 0.97, maxlter = 5
Spline	hmrR_MotionCorrectSpline	*p* = 0.99
SplineSG	hmrR_MotionCorrectionSplineSG	*p* = 0.99, FrameSize_Sec = 10
RLOEES	hmrR_MotionCorrectRLOEES	span = 0.02
Wavelet	hmrR_MotionCorrectWavelet	iqr = 1.5
CBSI	hmrR_MotionCorrectCBSI	On
Band-pass filter	hmrR_BandpassFilt	hpf = 0.01 Hz, lpf = 0.1 Hz
OD change	hmrR_OD2Conc	1.0 1.0 1.0
Average	hmrR_BlockAvg	−2.0 Sec 20.0 Sec

**Table 2 sensors-23-03979-t002:** Correction methods sorted by mean rank.

Method	Grand Mean Rank	Rank std
WCBSI	1.25	0.77
RLOESS	4.00	2.48
CBSI	4.13	2.13
Wavelet	4.25	2.27
SplineSG	4.94	2.24
Spline	5.44	1.46
tPCA	5.63	1.59
PCA	6.31	1.78
Uncorrected	8.63	0.81

**Table 3 sensors-23-03979-t003:** Correction methods sorted by standard deviation of rank.

Method	Rank Grand Mean	Rank std
WCBSI	1.25	0.77
Uncorrected	8.63	0.81
Spline	5.44	1.46
tPCA	5.63	1.59
PCA	6.31	1.78
CBSI	4.13	2.13
SplineSG	4.94	2.24
Wavelet	4.25	2.27
RLOESS	4.00	2.48

## Data Availability

The data is available in (https://www.doi.org/10.17605/OSF.IO/3A9Q6 (accessed on 24 March 2023)).

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
