# Peer review of "Improved Motion Artifact Correction in fNIRS Data by Combining Wavelet and Correlation-Based Signal Improvement"

_sensors, 2023, doi:10.3390/s23083979_

Round 1

Reviewer 1 Report

Reviewer

The following comments need to be considered:

Concern #1: The abstract section needs to be rewritten to be concentrated on the main contribution of research.

Concern #2: The methodology section should explain the used equipment to measure the required parameters. Also, the calibration methods during the measurement process.

Concern #3: The text and the explanation of the figures are very poor in the paper.

Concern #4: The flowchart, the steps, and the parameters of the design should be provided.

Concern #5: Statistical analysis of the design should be added such as standard deviation,

also convergence curve should be included.

Concern #6: The results are not well discussed, the curves are provided with a very brief

discussion. The section of results should have more details.

Concern #7: the references need to be updated.

Reviewer 2 Report

The authors describe an approach that combines wavelet and correlation-based signal improvement to minimize the detrimental effects of MAs. There are a few aspects that should be addressed before the paper may be processed further.

1. The authors claim that the described approach performs most favorably among the tested algorithms. This particularly important claim should be fully justified, with theoretical and practical arguments.

2. In this respect, the performance evaluation should be extended and fully explained. 

3. Furthermore, the literature review should better highlight the merits and potential drawbacks of the described approach compared with the models described in relevant existing papers.

4. The performance evaluation is performed on a rather weak system. It is unclear whether the system can scale to much larger and complex real-world use case architectures. The authors should discuss on the system's appropriateness for real-world use cases with arguments that are based on theory and experimental data.

5. The English language should be fully proofread and improved.

Round 2

Reviewer 1 Report

The paper is accepted in the present form

Author Response

We thank the reviewer for the constructive comments which helped to improve our work and for accepting our paper.

Reviewer 2 Report

Please provide sufficient background and all relevant references, the review section needs to be improved.
